# A Real-Time and Robust Neural Network Model for Low-Measurement-Rate Compressed-Sensing Image Reconstruction

**DOI:** 10.3390/e25121648

**Published:** 2023-12-12

**Authors:** Pengchao Chen, Huadong Song, Yanli Zeng, Xiaoting Guo, Chaoqing Tang

**Affiliations:** 1PipeChina Institute of Science and Technology, Langfang 065000, China; chenpc@pipechina.com.cn; 2SINOMACH Sensing Technology Co., Ltd., Shenyang 110043, China; 3China Belt and Road Joint Lab. on Measurement and Control Technology, School of Artificial Intelligence and Automation, Huazhong University of Science and Technology (HUST), No 1037 Luoyu Rd., Wuhan 430074, China

**Keywords:** compressed sensing, deep neural network, image reconstruction, low measurement rates

## Abstract

Compressed sensing (CS) is a popular data compression theory for many computer vision tasks, but the high reconstruction complexity for images prevents it from being used in many real-world applications. Existing end-to-end learning methods achieved real time sensing but lack theory guarantee for robust reconstruction results. This paper proposes a neural network called RootsNet, which integrates the CS mechanism into the network to prevent error propagation. So, RootsNet knows what will happen if some modules in the network go wrong. It also implements real-time and successfully reconstructed extremely low measurement rates that are impossible for traditional optimization-theory-based methods. For qualitative validation, RootsNet is implemented in two real-world measurement applications, i.e., a near-field microwave imaging system and a pipeline inspection system, where RootsNet easily saves 60% more measurement time and 95% more data compared with the state-of-the-art optimization-theory-based reconstruction methods. Without losing generality, comprehensive experiments are performed on general datasets, including evaluating the key components in RootsNet, the reconstruction uncertainty, quality, and efficiency. RootsNet has the best uncertainty performance and efficiency, and achieves the best reconstruction quality under super low-measurement rates.

## 1. Introduction

Compressed sensing (CS) is a promising technology which enables sub-Nyquist sampling, data compression, flexible measurement rate, etc. These benefits lead to ground-breaking achievements in many computer vision applications, e.g., image super-resolution [1], image de-noising [2], image registration [3], etc. Sub-Nyquist sampling and data compression lead to significant time-saving results in fault-detection applications like belts [4], bearing [5], composite materials [6], etc. It can also reduces the hardware requirements for monitoring applications like pipelines [7,8]. The flexible measurement rate enables robustness to partial data loss in harsh environments like nuclear sites. CS collects a linear mapping collection of a scene followed by a reconstruction/estimation process to obtain the final sensing data. Since the rigid proof of CS theory in 2006 [9], CS becomes hot research topic; however, this interest has gradually reduced in recent decades, leading to few applications—the high computational time for sparse estimation is a limitation in various applications, especially in imaging tasks. For example, the reconstruction time for traditional iterative optimization theory is usually unacceptably long. Some studies [10,11] report that it takes around 10 min to estimate an image of size 256 × 256 in block-by-block manner; a full-size reconstruction may take much longer. Deep learning [12] dates back to the previous century; it has developed quickly and has led to a huge amount of applications compared to CS, especially in recent years. To handle the problem of high computational time for sparse reconstruction in CS imaging, many scholars have been focused on bringing deep learning methods to CS estimation since 2016 [13].

Existing deep-learning-based CS estimation methods are usually classified into two categories [14,15]. The first category [13,16] uses a convolutional neural network (CNN) to setup end-to-end models from the CS measurement data to the final image. Early examples are ReconNet [13], SCSNet [17], CSNet+ [16], etc. CSNet+ jointly learns the measurement matrix and completed end-to-end reconstruction. SCSNet connects multiple neural networks that each one takes a measurement set from a measurement matrix for training. The purpose is to handle various of measurement rates (MRs) in real applications. This will incur huge amount of training parameters. Such end-to-end designs in the early stage of learning-based methods regards CS reconstruction as a black-box, which has poor reliability, i.e., the trained models are fitting to the given data rather than the CS reconstruction problem. This leads to trust problems in imaging contexts like MRI [18]. If the training dataset is small, or the training process is set under a specific measurement matrix, then the trained models will fail in different settings.

The second category is a newer trial direction, which tries to unpack traditional iterative optimization-theory-based methods like ADMM, ISTA, and IHT [19] into neural network layers. The core idea is to build a neural network module for each iteration step in the traditional optimization-based methods. This feature is advertised as an interpretable neural network in many studies on CS reconstruction. Zhang et al. [20] presented an iterative shrinkage thresholding algorithm (ISTA) and built up an ISTA-Net. Their updated versions are COAST [21] and ISTA-Net++ [22], which include the addition of a de-blocking strategy to eliminate the blocking artifacts. The alternating direction method of the multipliers algorithm (ADMM) led to the development of ADMM-Net [23], ADMM-CSNet [24], Co-robust-ADMM-Net [25], and, recently, GPX-ADMM [14]. AMP-Net [26] was a development of the traditional approximate message passing (AMP) method. Inspired by the primal–dual algorithm, a learned primal–dual network (LPD-Net) was built. J. Zhang et al. [27] derived a weighted *ℓ*${}_{1}$ minimization algorithm for CS reconstruction and built a deep unfolding network for it.

Besides the above two categories, some scholars have tried to create deeper links to traditional optimization methods. One representative idea is Pnp-ADMM [28], which uses neural network modules to model the error function, regularization, and Lagrange multiplier. This idea brings better interpretation but needs more iterations, which sacrifices the benefit of time efficiency for deep learning methods. Another idea is to build neural networks to find the support set during sparse reconstruction [29]; with the found support, the final image can be obtained through a further matrix reverse step. However, the support identification neural network module is a black-box; a wrong support set will lead to significantly different results.

The key considerations for reconstruction quality guarantee are the residual, noise, and block effects. Residual refers to the remaining difference between the explained part and the ground truth. Iteration steps, the measurement matrix, and sparse basis link directly to the reconstruction residual effect. Deep-learning-based methods are not as sensitive as optimization methods are to iteration steps but they are more sensitive to the sparse basis and the measurement matrix. Some deep-learning-based reconstruction works [30,31] also try to learn the measurement matrix with fewer measurements or better reconstruction quality. However, a learned measurement matrix fits to the trained data, thus losing generality. The block effect is incurred through decomposing the full image into small blocks for the purpose of increasing the speed. Some works [22,26] use a built-in or separate de-blocking module like BM3D [22] to remove the block effect. As for de-noising, traditional methods need prior information of noise like noise distribution in Bayesian CS [32,33], which needs a noise estimation step for reliable reconstruction, and the reconstruction algorithm is usually designed for only one kind of noise. Deep learning methods can learn this prior information from the training datasets and are able to handle multiple kinds of noise at the same time [26]. Deep learning also has the potential for task processing during reconstruction; this has recently emerged up in traditional CS reconstruction [34,35].

In summary, almost all the deep-learning-based CS estimation works can greatly improve the reconstruction speed. The quantity of neural network layers may impact the running time, but this impact is in a much lower order of magnitude than the leap from optimization-theory-based iterative methods to the deep learning regime. Deep learning also brings benefits to robust de-blocking and de-noising. However, none of the existing deep learning methods can match the generality and reliability that traditional methods attain, even if some deep learning methods are advertised as interpretable, i.e., through the uncertainty problem. The so called ‘interpretable’ nature of these approaches involves building neural network blocks to substitute the traditional iterative steps. One remaining key concern is ‘Can the end-user really trust the measurement results of current deep learning reconstruction?’ No current deep-learning-based methods in the measurement field can answer this question, which causes a gap between the good performance shown in papers and the true benefits being attained in application. This is more important in applications that require a high standard of reliability, e.g., medical imaging [36], infrastructure health diagnosis [37], etc.

In seeking truly trustworthy deep-learning-based CS estimation methods, this paper proposes a neural network called RootsNet. Instead of building neural network modules to substitute the iterative steps, this paper considers the issue from another perspective: can the reconstruction results be predictable if something goes wrong in the deep learning mode? The core problem of CS reconstruction is using the measurement matrix and sparse basis to find weightings of a small percentage of columns in sparse basis. For general sparse basis like DCT and DWT, each column has a clear physical meaning. For example, each column of DCT basis is corresponding to a single-frequency cosine wave. Fourier theory tells us that every signal can be decomposed to linear weighting of different frequency components. CS theory is based on a more general ground truth: every signal can be decomposed to linear weighting of different unit components which can be cosine waves, wavelets, or self-learned components. These unit components make up the sparse basis, and the weighting is referred to as ’sparse coefficient’. So, if a deep learning model can predict the sparse coefficients on a general sparse basis, then the reconstruction results are much more controllable, because each weighting has a clear physical meaning.

So, this paper proposes RootsNet, which consists a Feeder-root module to predict the sparse coefficients, and a Rootstock net module for residual processing and de-blocking. The measurement matrix, sparse basis, and the measurement data are transformed to root caps. The whole structure is similar to a root structure and named after it. The major contributions are:This paper proposes RootsNet for a small step toward truly trustworthy deep-learning-based CS image reconstruction. Instead of being a black-box as its counterparts are, RootsNet integrates the CS mechanism into the network to prevent error propagation. The error-injection test in Section 4.2.4 shows RootsNet is much more robust than its counterparts.RootsNet enables real-time reconstruction and supports different measurement rates in a single net for general measurement matrices. Section 4.2 validates this feature.RootsNet successfully reconstructs super-low measurement rates that are impossible for traditional optimization-theory-based methods. The qualitative evaluation on two real-world applications, presented in Section 4.1, shows this powerful ability. At least 60% of the measurement time is saved in one microwave testing system using the proposed method. The proposed method achieves extremely low measurement rates, which saved at least 95% of storage in one pipeline monitoring system. The quantitative evaluation, presented in Section 4.2.3, also validates this ability.

The rest of this paper is organized as follows: Section 2 remarks the key CS theory basis. The proposed RootsNet is introduced in Section 3. Detailed discussion and analysis are given in Section 4. Conclusions including limitations are summarized in Section 5.

## 2. Compressed Sensing Measurement Theory

CS theory is based on a signal that can always be decomposed to linear weighting of different unit components; the unit components can be cosine waves, wavelets, or self-learned components. If a small part of the weighting already occupies most of the total weighting power and the rest can be ignored, then the weightings become sparse coefficients. In a more rigid way, a vector signal $x\in R^{n\times 1}$ can be decomposed on a basis matrix $\Psi \in R^{n\times n}$ as x=Ψs, where s has only a small part of non-zero values. Then, the CS measurement data can be represented as:(1)y=Φx+ξ=ΦΨs+ξ=As+ξ
where $\Phi \in R^{m\times n}$, $y\in R^{m\times 1}$ are the measurement matrix and measured data, respectively. $A=\Phi \Psi \in R^{m\times n}$ is the multiply of measurement matrix and sparse basis. $\xi \in R^{m\times 1}$ is the sampling noise. The original signal x is compressed from *n* to *m* due to m≪n, leading to a data compression ratio of 1−m/n and the measurement rate is defined as m/n. This CS measurement process is an encoding process.

CS reconstruction is the decoding process, it reconstructs the sparse coefficients s with the measured data y and the known A:(2)$$\min_{s}{∥s∥}_{0}\mathrm{subject}\mathrm{to}{∥y-\mathrm{As}∥}_{2}\leq \varepsilon$$
where ${∥\cdot ∥}_{0}$ is zero-norm which means the number of non-zero elements in a vector, and ε is a residual tolerance. If the measurement matrix is designed to let A meet the restricted isometry property condition [38], then there are many theoretical guaranteed optimization solutions for this problem. Independent identically distributed Gaussian matrices [38] and 0/1-Bernoulli matrices [39] can be used as general measurement matrices.

So, Equation (Equation 2) means that, instead of finding the sparse coefficients of the target signal on a sparse basis directly, CS reconstruction converts to finding the weighting for each column of A to represent the measurement y. Traditional reconstruction methods light orthogonal matching pursuit (OMP) mainly based on convex optimization and greedy algorithms, which takes unacceptably long time in many applications. End-to-end deep learning reconstruction methods directly learn a mapping from the measured data to the final image. They obtain great efficiency gain but the results are not interpretable and have trustworthiness problem, i.e., if something goes wrong or out of anticipate, they do not know what will happen on the reconstructed images. The next section proposes RootsNet that use y and A to estimate s and x directly.
(3)$$\left[s,x\right]=f_{RootsNet}\left(y,A\right)$$

## 3. The Proposed Rootsnet

This section introduces the proposed RootsNet, which implements Equation (Equation 3) and answers what will happen if the key part of the network goes wrong while inheriting the benefits of deep-learning-based CS reconstruction methods.

### 3.1. Overall Structure of RootsNet

The overall structure of RootsNet is shown in Figure 1. RootsNet consists of three key parts, i.e., the root caps, the feeder root net module, and the rootstock net module. These three parts work together to form a structure like roots in the background, and the model is named after it. As a popular approach, one high-resolution image (pepper image in the bottom right corner of Figure 1) is decomposed into same size blocks firstly and reconstructing each block, respectively, or in parallel. Parallel reconstruction is much more efficient for hardware that supports tensor computing. A and the measurement data Y are compounded to many root caps; each root cap is the input of a single feeder root branch, $B_{n}$. A feeder root branch is responsible for predicting a single sparse coefficient value, $s_{n}$. With enough feeder root branches, the sparse coefficients can be fully predicted. Then, the image blocks can be obtained by simply imposing linear weighting on the sparse basis. The feeder root net module already substitute the full process of traditional optimization algorithms. Lastly, a rootstock net module is imposed on the block reconstruction results to handle the estimation error, the block effect, and the noise.

### 3.2. Key Modules in RootsNet

#### 3.2.1. Root Caps

Root caps are the input of each feeder root net module. As is shown in Figure 1, an image is decomposed to *b* blocks in same size as L×L, then reshaping to a matrix $X\in R^{n\times b}$. With a measurement matrix $\Phi \in R^{m\times n}$, the measured dataset $Y\in R^{m\times b}$ is compressed to m/n of X. CS reconstruction is finding the weighting for each column of A (denote as $A_{i}$, *i* = 1, 2, 3, …, *n*.) to represent the measurement vector y, so $A_{i}$ and each column of Y (denote as $Y_{j}$, *j* = 1, 2, 3, …, *b*.) are compounded to be root caps, the corresponding feeder root branch will output the weighting for $A_{i}$. Each feeder root branch takes only one root cap as input and outputs one sparse coefficient. To fit with the input port of feeder root net, $A_{i}$ and $Y_{j}$ are reshaped to two images of size $\sqrt{m}\times \sqrt{m}$ and then simply stack together. The size of root caps requires *m* to be the square of integers, which results in discontinuous for the supported MRs. To enable support for arbitrary sampling rate, root caps pads zeros to obtain a fixed size virtual m that meet the condition. In real applications, the virtual *m* is recommended to be the largest possible MR (usually to be around 0.3), so that a single feed root branch can fit all measurement rates cases.

#### 3.2.2. The Feeder Root Net Module

Each feeder root branch is responsible for reconstructing a sparse coefficient. The structure of a single feeder root branch is shown in Figure 2. Two convolution (Conv_1 and Conv_2) and pooling layers (Pooling_1 and Pooling_2) extract features from root caps firstly. Max pooling operation and nonlinear activation function relu() are used. Conv_1 takes two channels of input to fit with the structure of root caps. After feature extraction, all features are flattened before sending to a dropout layer. This dropout layer can effectively overcome overfitting and make the learned feature more robust. Finally, a fully connected (FC) layer combines all the learned feature together to a single value, which is the target sparse coefficient.

Feeder root simulates the correlation calculation steps in traditional matching pursuit (MP) algorithms for CS reconstruction. In a typical MP algorithm like orthogonal matching pursuit (OMP) [40], the measurement y will calculate correlation for every $A_{i}$; the corresponding sparse coefficient for $A_{i}$ can be regarded as the correlation coefficient between y and $A_{i}$. $A_{i}$ and y are used to form a root cap, so the feeder root net modules substitute the correlation calculation in traditional MP algorithms.

The feeder root net is scalable and distributable. Sparse representation is the theory basis for CS, i.e., only a small part of the sparse coefficients is significant, and the rest of the values can be ignored. For this reason, for an image of flattened size n×1, the number of feeder root branches can be much lower than *n* by only predicting a part of the most significant sparse coefficients. The contribution of later branches to image quality is less significant, which means that the branch number is scalable. Due to the independence between each branch and the strict one-to-one mapping between a feeder root branch and a sparse coefficient, the feeder root net can be distributable. This distributable feature is helpful for resource-limited devices like wireless sensor networks for structural health monitoring [41].

#### 3.2.3. The Rootstock Net Module

Ideally, the feed root nets already fully substitute the traditional CS reconstruction results, because all sparse coefficients can be obtained. In fact, there is some reconstruction noise for the feeder roots net outputs due to wrong prediction and the limited number of feeder root branches (these two factors also lead to the block effect), or the sampling data already polluted by noise.

To further improve the reconstruction quality, a rootstock net that takes all block reconstruction image as input is used; the structure is given in Figure 2b. It is a fully convolutional network which has different kernel sizes (denote as k()), padding numbers (denote as p()), and kernel numbers (denote as conv_()). The first three layers use relu() as activation function, while the last three layers use tanh(). The final crop operation together with the padding in each layer can remove the edge effect in the final image.

### 3.3. The Underlying Information Theory for RootsNet

The measurement data y comprise the weighted summation of some columns of A, in theory. The task for reconstruction algorithms is to determine the sparse column positions and the corresponding weighting values. Traditional greedy algorithms calculate the correlation values between each column of A and y, because if one column of A belongs to the components of y, it should have high correlation value, then removes the contribution of this column in y to obtain the residual $\hat{y}$, followed by another round to find the next column. So, the root caps module uses y and each column of A as input, which is the same for greedy algorithms. Then, the feeder root net module uses parallel branches to predict the corresponding weight values directly. So, even if some branches go wrong, they will not influence other branches. This is one fundamental difference from existing deep learning methods, where a tiny defect in the network structure or prediction process may leads to totally different results.

Wrong predictions from the feeder root net can be modeled as adding a noise on the ground truth sparse values. For a general sparse basis like DCT, the wrong prediction only modifies the corresponding frequency weighting. If the frequency is in the middle–high frequency range, the overall visual quality of the image will not be significantly different. More importantly, for a general sparse basis like DCT, the sorted sparse values are in a similar sequence, which can be used as a priori information to judge how trustworthy a prediction result is. For example, an outlier is a highly unreliable prediction. On the other hand, a wrong prediction can be corrected using the following rootstock net module.

### 3.4. Training Methods

RootsNet can be trained with open-source datasets like ImageNet; this paper used the BSDS500 [42] dataset for training. There are 500 colorful visual images in BSDS500. All the images were converted to grayscale and sized to 256 × 256. The feeder roots net and rootstock net were trained separately. Feeder roots branches can be trained individually or together. All neural network layers were implemented with Paddlepaddle 2.3.0 https://github.com/PaddlePaddle/Paddle (accessed on 18 October 2023). The training process was implemented on a desktop computer with an Intel Core i9-10900K CPU and an NVIDIA GeForce RTX3090 GPU with Python.

This paper firstly trained the feeder root branches individually; a cost function in Equation (Equation 4) was designed, where $s_{p}$ and $s_{g}$ are the predicted and ground truth sparse coefficients, respectively, and ς is a random, small-enough value to prevent the denominator to be zero. This cost function combines the absolute error and relative error, which gives better results than the commonly used mean square error function in our test. $s_{g}$ is obtained by setting the block size *L* as 32 and decomposing each block on sparse basis. Both DCT and Haar wavelets are considered as the sparse basis. 0/1-Bernoulli and random Gaussian matrices are tested as measurement matrices. The virtual *m* is set as 361, which corresponds to a max MR of mmnn≈0.353. More measurements will counteract the compressing benefit of CS for most applications. Users can set it as other values according to their needs. A predefined measurement matrix is easier to implement in real applications, so this paper chooses seven predefined MRs from 0.05 to 0.35 with a step-size of 0.05. All kernel sizes are set as 3, the stride step-size for convolution layer and max-pooling layer are set as 1 and 2, respectively. The convolution kernel numbers are set as 64 for Conv_1 and Conv_2. The issue of deciding a minimum kernel number remains a challenge for researchers across the field of deep learning. The learning rate is initialed as 0.01 and drops to 0.00001 through polynomial decay during 3000 epochs of training. The dropout rate is set as 0.3.
(4)$$ℑ\left(s_{p},s_{g}\right)=\frac{{\left(s_{p}-s_{g}\right)}^{2}}{\left|s_{g}+\varsigma \right|}$$

Secondly, the rootstock net takes the output of feeder roots net as input for training. The minimum reconstruction unit for rootstock net is a full image rather than image blocks. During training, the loss metric is set to be structural similarity (SSIM [43]) in Equation (Equation 5) between the model prediction and the ground truth, where $\mu_{J}$ and $\mu_{K}$ are the mean intensities of image *J* and *K*, respectively. $\sigma_{J}$ and $\sigma_{K}$ are the standard deviations of images *J* and *K*, respectively. $\sigma_{JK}$ is the covariance of images *J* and *K*. $\varsigma_{1}$ and $\varsigma_{2}$ are two constant small values that prevent the denominator from being zero. The feeder root net predictions under different branch numbers and MR are used for training. To obtain a de-noising ability for the rootstock net, the Gaussian noise and salt and pepper noise are added in the feeder root net output for training. The other training settings are the same as those used for the feeder roots training.
(5)$$\mathrm{SSIM}\left(J,K\right)=\frac{\left(2\mu_{J}\mu_{K}+\varsigma_{1}\right)\left(2\sigma_{JK}+\varsigma_{2}\right)}{\left({\mu_{J}}^{2}+{\mu_{K}}^{2}+\varsigma_{1}\right)\left({\sigma_{J}}^{2}+{\sigma_{K}}^{2}+\varsigma_{2}\right)}$$

## 4. Experimental Results

### 4.1. Qualitative Evaluation in Real-World Applications for Low Measurement Rates Reconstruction

The proposed RootsNet is firstly tested qualitatively on two real image measurement applications, i.e., near-field microwave imaging of carbon fiber reinforced polymer (CFRP) and magnetic flux leakage pipeline inspection gauge for oil and gas pipeline inspection, which aims to show the ability of reconstructing under a high data compression ratio. There are no deep learning reconstruction methods implemented in these two applications in the literature yet, only iteration-based methods. Besides existing iterative methods, two typical deep learning CS reconstruction methods are also implemented here to show the performance gain on super-low measurement rates, i.e., ReconNet [13] and AMP-Net [26], because they are representative methods for existing deep-learning-based CS estimation—the end-to-end category—and another category of developing traditional optimization theories. All the reconstruction algorithms are implemented on a desktop computer with an Intel Core i9-10900K CPU with Python, because a trained neural network does not require a powerful GPU to run. Deep learning training and reasoning are implemented on Paddlepaddle 2.3.0.

#### 4.1.1. Application in Near-Field Microwave Imaging

CFRP materials are widely used in the aerospace industry due to their good weight-to-strength ratio. Invisible impact damages on CFRP greatly influence the strength and lead to safety risks. Near-field microwave imaging is one common technology that is used for invisible impact damage detection, but current methods only use raster scan or traditional iterative reconstruction methods [34,35], which take hours to perform detection.

Considering the real-time reconstruction ability of RootsNet, the same measurement settings were used as those used in [35] for impact damage detection with the system that is shown in Figure 3. Five specimens with impact energy from 2J to 10J, respectively, are used. An extremely low measurement rate of 0.05 is used, which means that only 5% of locations in the whole scanning area are scanned. Figure 4 shows the measurement results. The raster scan data and CS scan data under the measurement rate of 0.05 have a low correlation with the ground truth. The state-of-the-art method in Figure 4c has much lower reconstruction quality than RootsNet, and the reconstruction for the 8J specimen even failed due to the low measurement rates. The proposed RootsNet also has better reconstruction quality than ReconNet and AMP-Net-9BM, because only RootsNet is designed for the sparse-coefficient-level reconstruction and it enables error correction on the sparse coefficients; other deep learning methods perform image-level reconstruction. A low measurement rate like 0.05 in this test brings reconstruction errors to the ground truth; these errors will lead to artifacts on the final image if there is no error correction scheme.

The normalized time under the measurement rate of 0.05 to obtain the corresponding results is shown in the bottom part of Figure 4, where the time for 100% raster scan is set as the normalized baseline. The total time consumption shown in Figure 4c–f consists of the scanning time to obtain the input data and the reconstruction time. During the scanning process, the scanner needs to switch between different locations, and each location needs a constant delay time to prevent scanner vibrations and to perform the measurements. The typical delay time is 0.1–0.5 s for each scanning location and it is set as 0.1 in this test, so the scanning takes much more time than the reconstruction. The reconstruction time is linked to the computational resources utilized; iterative reconstruction methods take much more time than a trained deep learning model, especially when using a not-so-powerful CPU. Some existing experiments [35] indicated that the traditional OMP needs around 0.25–0.3 measurement rate to achieve more than 0.95 reconstruction quality. However, a large measurement rate leads to great increases in the scanning time and the reconstruction time with the OMP. For a traditional OMP to obtain more than 0.95 reconstruction quality, the normalized time is 35 for a measurement rate of 0.3—this compared with just 1 for a measurement rate of 0.05. The deep learning methods generate the reconstruction results almost immediately, so the normalized reconstruction time is around 0, compared to 1 for the OMP in Figure 4c. The total time consumption for OMP under a measurement rate of 0.3 is 65, including 30 for scanning and 35 for reconstruction. So, to obtain an image quality that meets the ground truth, the CS scan using RootsNet saves 95% and 60% of the measurement time in comparison with the raster scan and the state-of-the-art OMP, respectively.

#### 4.1.2. Application in Pipeline Inspection Robot

Oil and gas are the main sources of fuel in today’s industry, and pipelines are the most efficient way of transporting large amounts of oil and gas. Defects in pipelines lead to leakages or explosions that greatly impact human safety and ecosystems. Magnetic flux leakage (MFL) measurements using pipeline inspection gauges (PIGs) are the most popular method of detecting pipeline defects. PIG with MFL sensors will be sent into the pipeline and collect MFL data while moving along the pipe. One challenge is that in-service oil and gas pipelines are usually thousands of kilometers long, requiring unacceptable amounts of storage for reasonable resolutions to be attained. So, RootsNet can be implemented to address this challenge, using a 168-channel PIG device from SINOMARCH Sensing Co., Ltd. (Beijing, China) (shown in Figure 5) and a real in-service oil pipeline operated by PipeChina.

Traditionally, MFL sensor samples operate evenly in the time domain with a constant time interval. As is shown in Figure 5, the 168 channels of MFL sensor modules are shut down randomly for 95% of the time slots in RootsNet, which are controlled by the 0/1 Bernoulli measurement matrix. Each measurement piece has 1024 time slots and is controlled by the same measurement matrix. So, the wake-up time slot is only 1024 × 5% = 51 in each measurement piece, and CS measurement data comprise an under-sampling version of the traditional full-time wake-up sampling. Figure 6 shows one measurement piece of two MFL sensor channels. The traditional OMP reconstruction estimates poorly under such low measurement rates, but the proposed RootsNet almost perfectly reconstructed it. Figure 7 and Figure 8 show two pieces of measurement for the whole 168 MFL sensor channels, respectively. The 2D correlation coefficients are given on the northeast corner of each measurement result section. The OMP failed to reconstruct the pieces two due to the measurement rate being too low. RootsNet reconstructed this well, with very high quality. So, RootsNet saved at least 95% of storage and sensor power consumption in this application. Similar to the results attained in near-field microwave imaging, ReconNet and AMP-Net-9BM obtained poorer results than the proposed RootsNet under this super-low measurement rate.

### 4.2. Quantitative Evaluation on SET11

Without losing generality, this part completed testing on a popular dataset, SET11 [10], which is commonly used to evaluate CS reconstruction algorithms. Some key components for the proposed RootsNet from the CS perspective are quantitatively tested on a popular general dataset, e.g., determining the influence of feeder root branch number and how the sparse basis and measurement rate in CS influences the network performance. Computational memory efficiency is not tested because it is not a key point in this paper, but the distributable feature of RootsNet does have advantages in this area.

#### 4.2.1. The Influence of Sparse Basis and Roostock Net Module

This section qualitatively evaluates the ability of the feeder root net module to reconstruct the sparse coefficients firstly. Figure 9 shows two images that were recovered from the reconstructed sparse coefficients by the feeder root net and the ground truth in SET11; zoomed-in views of figureprint are shown in Figure 10. Ground truth coefficients are set by ensuring that the rest of the coefficients beyond the feeder root branch have a number of zero. Only 32 feeder root branches show the general texture for DCT basis, while it needs at least 128 feeder root branches for DWT. The floor number for the DWT basis is decided through wavelet decomposition orders: this is 128 for 3-order decomposition and 64 for 4-order decomposition. The red square in Figure 10 highlights the block effect, i.e., a clear edge line between reconstruction blocks. The block effect under the 256 feeder root branches is more serious for DCT, but DWT incurs a small block effect itself by using the Haar wavelet. Generally, the DCT has a better visual quality because cosine waves are smoother than Haar waves, as shown in Figure 11c. Using other wavelets like fk4 and sym2 can lead to smoother images than using the Haar wavelet. More feeder root branches lead to better texture detail due to more detail wave components being used, such as higher frequency cosine waves in Figure 11a,b. The visual quality of the ground truth images does not show significant differences after a certain branch number for both DWT and DCT bases; this means setting the feeder root branch number as 384 or 512 in this dataset. This is a good trade-off between reconstruction quality and efficiency.

Secondly, to handle the block effect and estimation noise, the feeder root net output are gathering to the rootstock net. Figure 12 shows the refine performance of the rootstock net module under some worst cases. The reconstruction quality improves greatly, even better than the baseline in some cases as is highlighted in the red circle. Rootstock net can remove the small block effect incurred by Haar wavelet as is shown in Figure 12. This simple test shows that RootsNet can handle the block effect, the estimation noise, and the error well.

#### 4.2.2. The Influence of Feeder Root Branch Number on RootsNet

To obtain quantitative results for the proposed RootsNet, the two popular evaluation metric peak signal-to-noise ratio (PSNR [13]) and SSIM are used. Figure 13 shows the average results for different branch number on SET11. More branches lead to better reconstruction quality because more components in the sparse basis are considered. DCT basis obtains better quality than DWT basis, which coincides with the qualitative results in Figure 10. Vertically, rootstock net greatly improves the quality from feeder root net output. Under 512 feeder root branches, the block effect of the feeder root net output can be ignored. The results of Figure 13 did not include noise images as the test input. Under the noise case, the PSNR and SSIM of RootsNet output can be better than the baseline shown in these figures. Because noise is usually in the high-frequency part of sparse coefficients, the feeder root net only reconstructs the major components in the low- and middle-frequency parts, which works like a de-noising filter. So, if using noisy images as input and the baseline, the RootsNet output is a denoised version of the baseline. The PSNR, which indicates the signal-to-noise ratio of an image should be higher. SSIM evaluates the similarity between two images from luminance, contrast, and structure point of view. When calculating SSIM using the noise-free images as the ground truth, RootsNet output can have better SSIM than the baseline. On the other hand, twisting the structure of rootstock net may improve the model performance.

#### 4.2.3. The Influence of Measurement Rates on RootsNet

Measurement rate is a key parameter in CS that shows data compression ratio. Figure 14 shows the reconstruction results under 256 branches for different MRs. Generally, higher MRs lead to better reconstruction results but it is not so sensitive as branch numbers. An MR of 0.05 already shows the general texture, both for DCT and DWT basis. Under same MR, the quality of DCT basis is better than DWT basis. Figure 15 shows the quantitative results for different MRs with 256 branches. Both PSNR and SSIM almost linearly increase with measurement rate. Rootstock net also shows great improvements in reconstruction quality.

#### 4.2.4. Evaluation of Robustness

This section compares the robustness performance of the RootsNet to two typical deep learning CS reconstruction methods—CSNet+ [16] and AMP-Net [26]. They are representative methods for existing deep-learning-based CS estimation, i.e., the end-to-end category and another category, in developing the traditional optimization theory. The main reliability concern for this paper framed in the following question: what will happen to the measurement results (with the reconstruction image as a case study) if some modules generate unexpected results? This happens when the measurement target has new features that have not been learned by the neural network or the low measurement rates, or they occur as a result of a software defect or a memory error. So, a robustness test is performed by manually stimulating the middle results of the network blocks. CSNet+ jointly learns the measurement matrix and reconstruction process, the reconstruction process consists of an initial module and multiple convolution modules. AMP-Net also has multiple modules that correspond to traditional iteration processes. Of the results, 1% and 10% for the first module of all the testing methods were manually set to a constant value (e.g., 0.3) for error injection. The measurement rate was set as 0.25. The results for all the testing methods are shown in each column of Figure 16. CSNet+ and AMP-Net incur artifacts quickly even with a 1% error injection, which means a tiny measurement or estimation noise can easily lead to uncertain results. RootsNet shows much better results even under 10% error injection, which reduces the uncertainty caused by the measurement noise and the estimation error. RootsNet obtains better performance, because only RootsNet integrates the CS mechanism in the network, i.e., it is designed for sparse-coefficient-level reconstruction through the root caps module and the feeder root net module, and it enables error correction on the sparse coefficients through the rootstock net module; other deep learning methods perform image-level reconstruction.

#### 4.2.5. Evaluation of Reconstruction Time

Popular traditional CS reconstruction algorithms include greedy algorithms (e.g., OMP, IHT [46]) and convex optimization methods (e.g., SpaRSA [47]); they reconstruct the CS measurement iteratively before reaching a predefined stop threshold, and there may be some hyper-parameters that are usually set empirically. Deep-learning-based CS reconstruction methods (e.g., ReconNet, ISTA-Net+) can greatly reduce the reconstruction time. This part records the average reconstruction time for both the optimization-theory-based methods and the deep-learning-based methods to show the efficiency of the proposed RootsNet approach. All reconstruction methods are implemented on a desktop computer with an Intel Core i9-10900K CPU with Python. The training process of deep learning methods is implemented on an NVIDIA GeForce RTX3090 GPU, and the trained networks only run on the CPU. The distributed implementation of RootsNet is implemented using parallel processing on the same computer. So, all reconstruction methods use the same hardware and software platforms. As is shown in Table 1, block reconstruction saves significant time. The time taken for all optimization-theory-based methods increases greatly with greater MRs from just several seconds to hundreds of seconds. The reconstruction time for deep learning methods are not sensitive to MRs, which almost keeps the same for different MRs because the number of forward parameters is the same for low and high MRs. The proposed RootsNet supports distributed reconstruction, which has the least parameters for a single feeder root branch. So, a state-of-the-art reconstruction time is achieved, which is suitable for real-time measurement applications. Generally, deep learning methods are at least 100 times faster than traditional optimization-theory-based methods.

#### 4.2.6. Evaluation of Reconstruction Quality

This section compares the proposed RootsNet with some state-of-the art methods on SET11; the PSNR in dB and SSIM are shown in Table 2. The test results of some deep learning methods come from recently published papers [14,26]. All tests in this paper use an MR range of 0.01–0.35, because higher measurement rates counteract the compressing benefit of CS, so five typical MRs are chosen in Table 2. The best and worst results are highlighted in red and green, respectively. A total of 512 feeder root branches are used in this test. Generally, deep learning methods have better reconstruction quality because the neural network models learned some a priori information during training. Block reconstruction leads to poorer performance due to the incurred block effect. Among all methods, the proposed RootsNet is sub-optimal in reconstruction quality for measurement rates above 0.1. RootsNet obtains the best performance in super-low MRs—those below 0.1—because only RootsNet is designed for sparse-coefficient-level reconstruction and it enables error correction through the rootstock net; other deep learning methods perform image-level reconstruction. A low measurement rate like 0.05 in this test brings reconstruction errors to the ground truth; these errors lead to the presence of artifacts in the final image if there is no error correction scheme. This study did not give attention to the effort of improving the reconstruction quality. For example, the structure of the rootstock net is very simple; there is more twist, and adding a cross-like layer connection may be helpful in improving the reconstruction quality.

## 5. Conclusions

In dealing with the significant time consumption and low reliability encountered in current CS image reconstruction, this paper proposes RootsNet. RootsNet outperforms current deep learning CS reconstruction methods in the following aspects: Firstly, by mapping the network structure to sparse coefficients, RootsNet prevented error propagation. This unique feature ensures that it can achieve a good performance under super-low measurement rates, where other deep learning CS reconstruction methods suffer from error propagation. The qualitative evaluation sections, error injection test, and the data in Table 2 demonstrate this advantage. The applications in a near-field microwave imaging system and a pipeline inspection system show that a CS with RootsNet easily saves 60% of measurement time and 95% of data in comparison with the state-of-the-art measurement techniques in the field, respectively. RootsNet also obtains better results in comparison with other CS deep learning reconstruction methods under a measurement rate of 0.05 in these two applications. Without losing generality, quantitative tests on SET11 show RootsNet achieves the state-of-the-art reconstruction quality for super-low measurement rates—below 0.1. Secondly, the proposed RootsNet enables distributed implementation. This unique feature makes it suitable for resource-limited applications like wireless sensor networks, where it is impossible for current iterative CS reconstruction methods and deep-learning-based methods. This feature also ensures that the proposed RootsNet approach can obtain the best efficiency for distributed implementation. Although it was only tested on a two-dimensional image dataset, the proposed method also works for one-dimensional and high-dimensional signals through reshaping them into images.

RootsNet is just a small step toward truly trustworthy deep-learning-based CS image reconstruction. The reliability is still lower than that of optimization-theory-based methods, i.e., the feeder root net module has good reliability, but the rootstock net still leads to measurement uncertainty. There is also some room to improve the reconstruction quality. For example, the structure of the rootstock net is very simple; there is an addition of a twist-like addition in the cross-layer connection that may be helpful in improving the reconstruction quality. A cost function that integrated more physical mechanisms of the CS may supervise the network to obtain better performance. On the other hand, no single deep learning method has the same level of generality as the traditional optimization-theory-based methods currently do, e.g., RootsNet works for predefined measurement matrices but cannot deal with arbitrary measurement matrices and sparse basis.

## Figures and Tables

**Figure 1 entropy-25-01648-f001:**
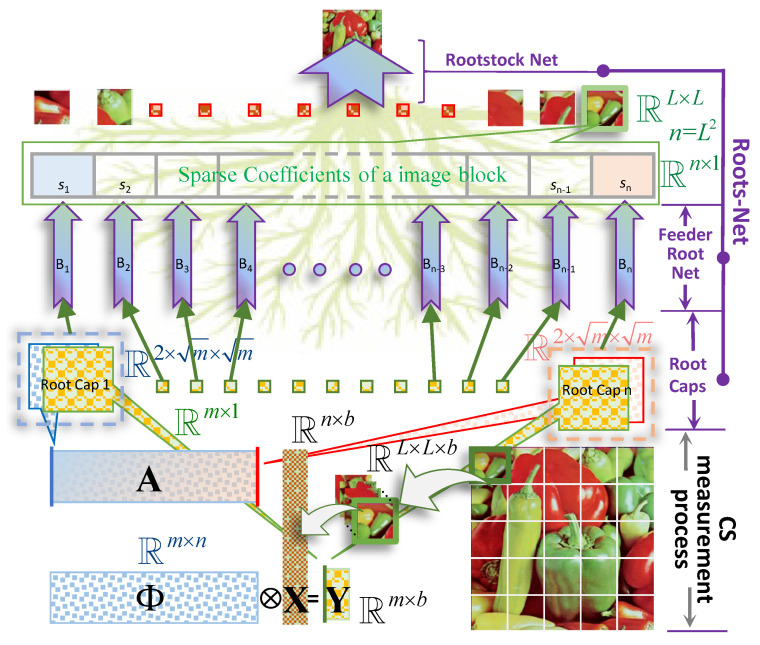
The compressed sensing measurement process and the overall structure of RootsNet. The target image is reshaped to *b* blocks with size L×L and reshaped to a n×b matrix latter as the target signal X, where $n=L^{2}$. RootsNet consists of root caps, feeder root net module, and rootstock net module as is annotated in purple text. Each root cap takes one column from Y and A, respectively, as input. The feeder root net consists of many branches that are denoted as $B_{1}$–$B_{n}$, each branch takes one root cap as input and outputs one sparse coefficient $s_{n}$. Finally, all reconstruction blocks are used as input to obtain the final reconstructed image through rootstock net. More details on each module are given in the next subsection.

**Figure 2 entropy-25-01648-f002:**
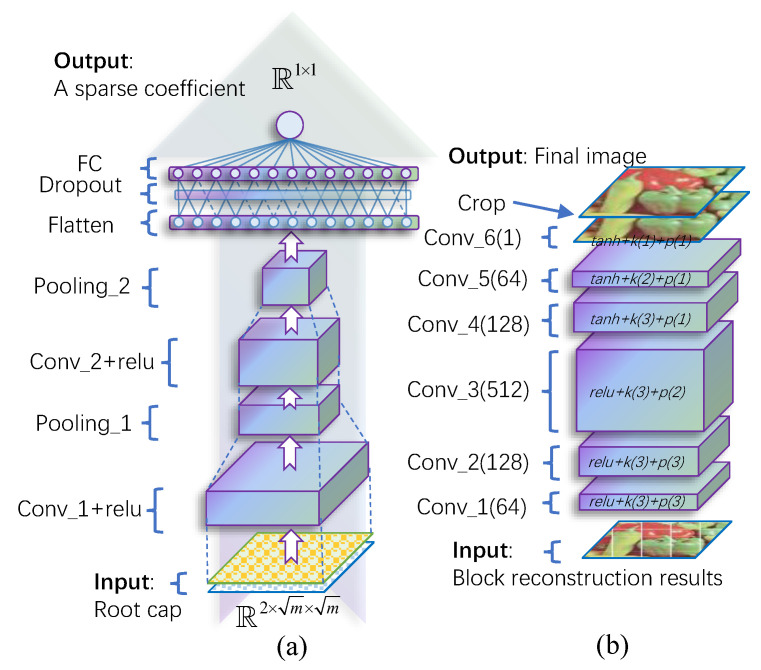
Network structure of (**a**) a single feeder root branch and (**b**) the rootstock module.

**Figure 3 entropy-25-01648-f003:**
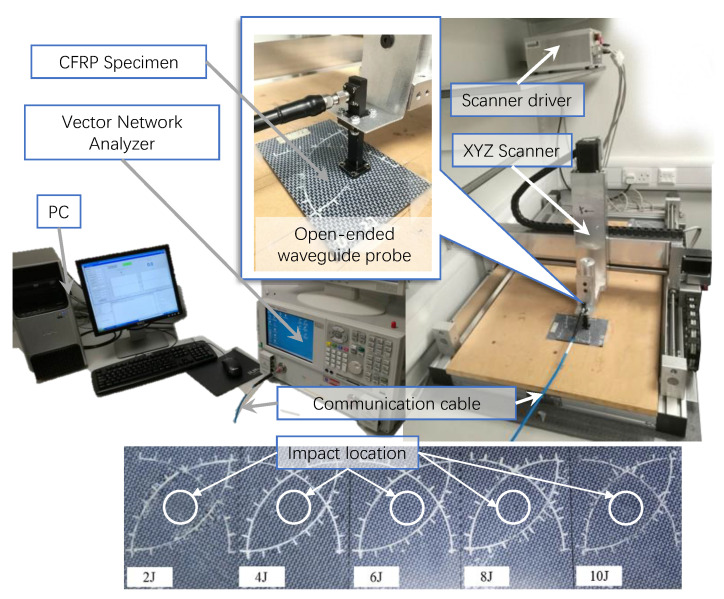
The image measurement system for invisible CFRP impact damage detection.

**Figure 4 entropy-25-01648-f004:**
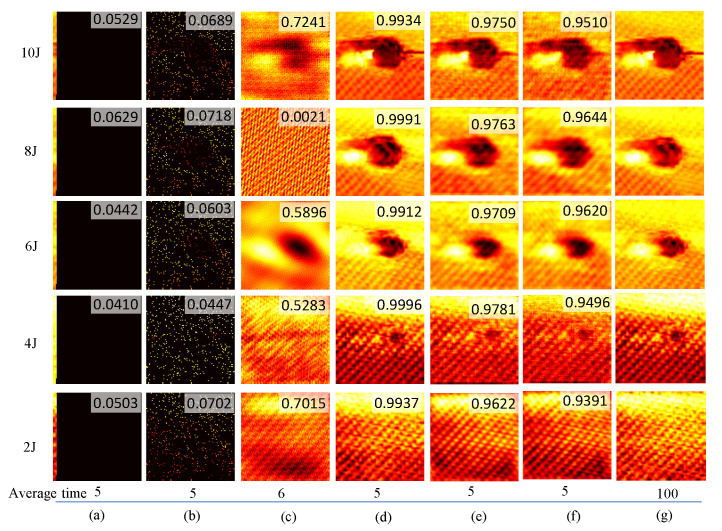
Measurement results of different methods under 0.05 of measurement rate (95% of data compression ratio). (**a**) Raster scan; (**b**) CS scan; (**c**) OMP reconstruction; (**d**) RootsNet reconstruction; (**e**) AMP-Net-9BM reconstruction; (**f**) ReconNet reconstruction; (**g**) The ground truth by 100% of raster scan. The built-in decimals are the 2D correlation coefficients between each measurement result and the corresponding ground truth image in column (**g**). The average normalized time use ground truth as the baseline and set it as 100.

**Figure 5 entropy-25-01648-f005:**
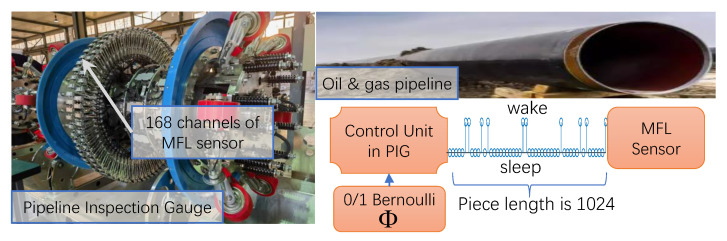
The MFL measurement system for oil and gas pipeline inspection.

**Figure 6 entropy-25-01648-f006:**
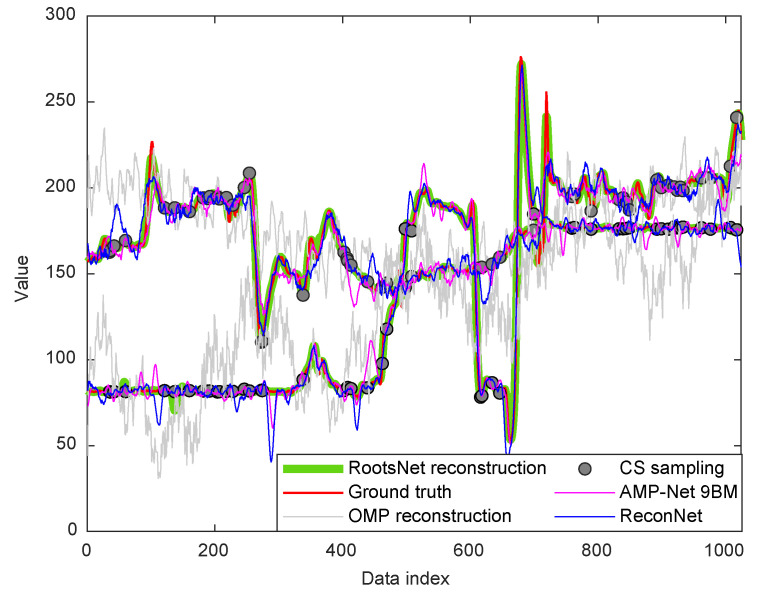
Examples of two measurement channels of one measurement piece under a measurement rate of 0.05 for pipeline. The ground truth is the traditional all-time wake-up measurement.

**Figure 7 entropy-25-01648-f007:**
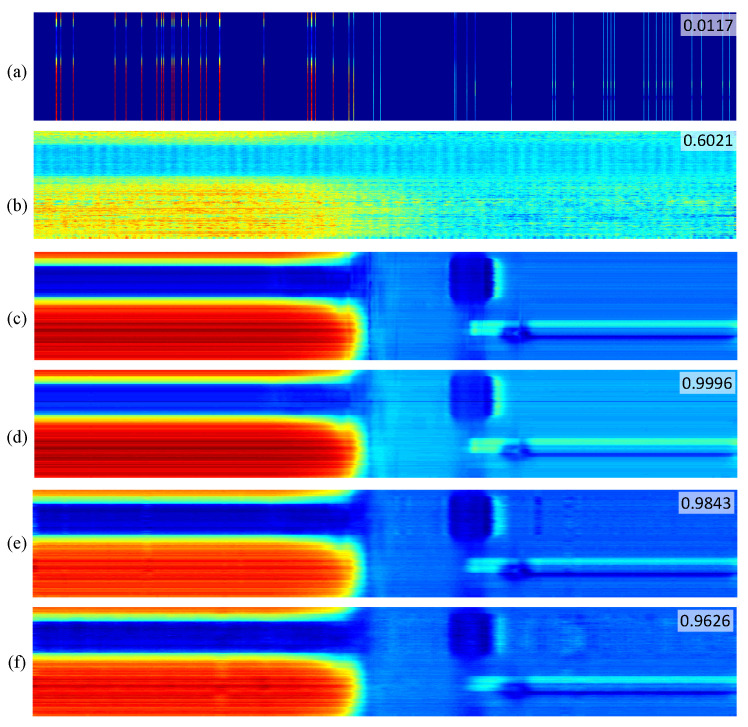
Measurement results of piece one under a measurement rate of 0.05 for the pipeline. (**a**) CS sampling data; (**b**) OMP reconstruction; (**c**) ground truth by full-time wake-up sampling; (**d**) RootsNet reconstruction; (**e**) AMP-Net-9BM reconstruction; (**f**) ReconNet reconstruction. Each row is one measurement channel. The proposed method successfully reconstructed a super-low measurement rate of 0.05, while the traditional OMP algorithm failed.

**Figure 8 entropy-25-01648-f008:**
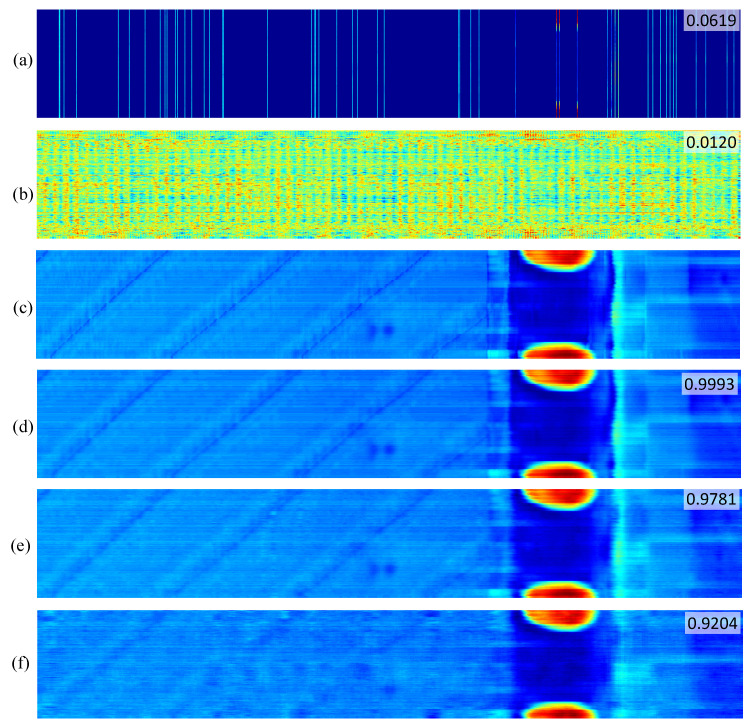
Measurement results of piece two under the measurement rate of 0.05 for pipeline. (**a**) CS sampling data; (**b**) OMP reconstruction; (**c**) ground truth by full-time wake-up sampling; (**d**) RootsNet reconstruction; (**e**) AMP-Net-9BM reconstruction; (**f**) ReconNet reconstruction. The proposed method successfully reconstructed a super-low measurement rate of 0.05 while traditional OMP algorithm failed.

**Figure 9 entropy-25-01648-f009:**
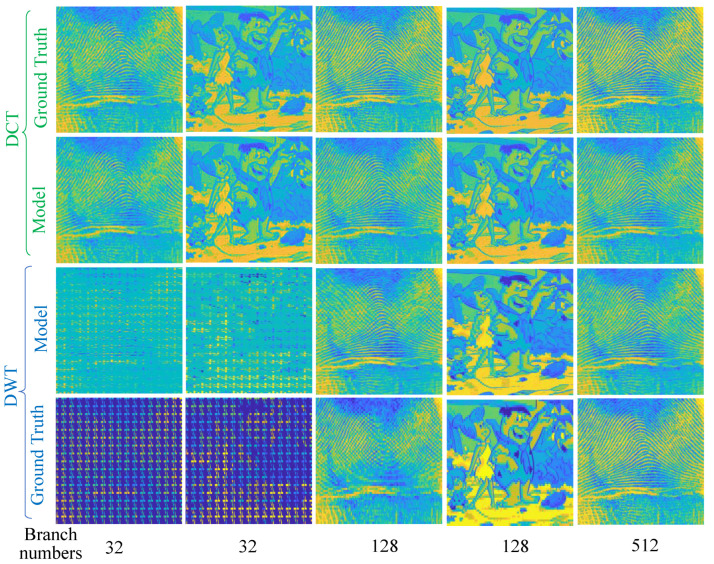
figureprint (odd columns) and flinstones (even columns) images recovered from the ground truth sparse coefficients and predicted coefficients of the feeder root net (denoted as Model in the figure) under different branch numbers and sparse bases. MR = 0.25.

**Figure 10 entropy-25-01648-f010:**
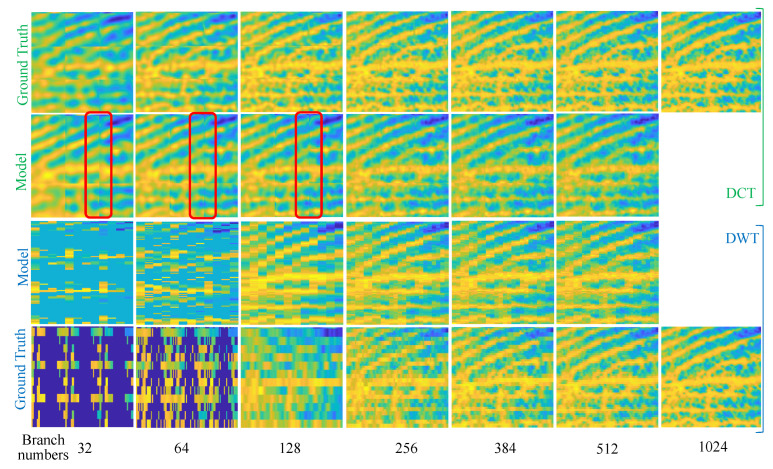
Zoomed -in view of figureprint in Figure 9.

**Figure 11 entropy-25-01648-f011:**
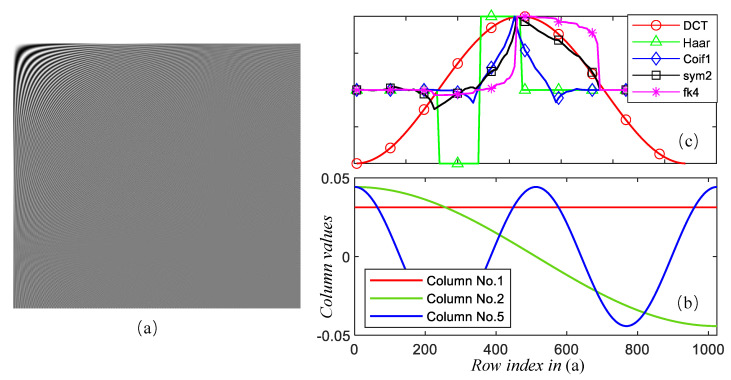
Waveform of different sparse bases. (**a**) a full DCT basis; (**b**) some columns in DCT basis; (**c**) cosine and other wavelets [44,45].

**Figure 12 entropy-25-01648-f012:**
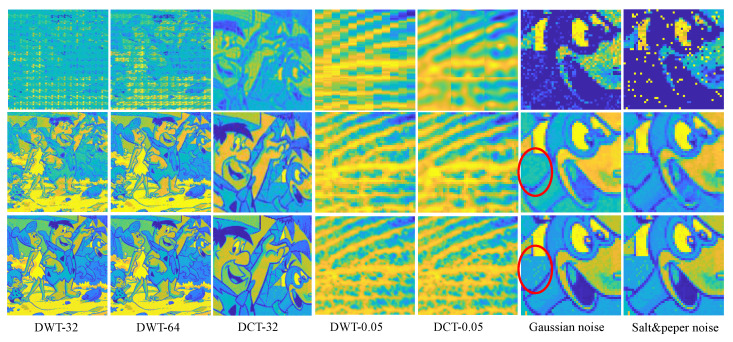
Refined performance for the rootstock net module under some worst case scenarios. Rows from the top to bottom are the feeder root net module output, the rootstock net module output, and the ground truth image, respectively. The number behind ‘-’ represents branch number for integers or MR for float decimals.

**Figure 13 entropy-25-01648-f013:**
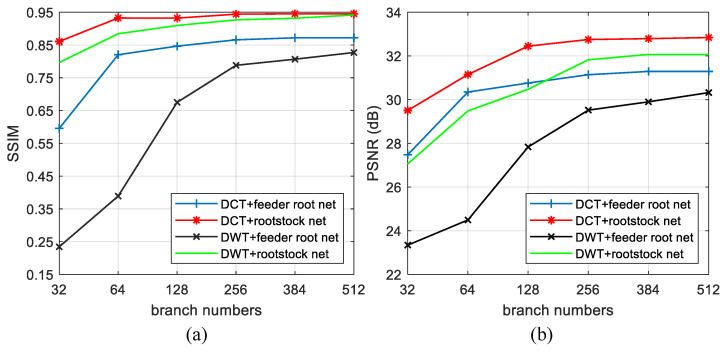
The influence of branch number on average reconstruction quality in SET11 by (**a**) SSIM and (**b**) PSNR. MR = 0.25.

**Figure 14 entropy-25-01648-f014:**
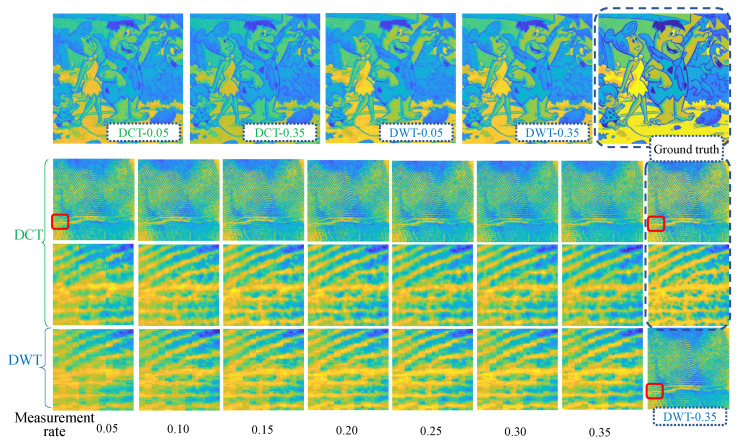
Reconstruction results under 256 feeder root branches for different measurement rates. The red square shows the zoom-in location.

**Figure 15 entropy-25-01648-f015:**
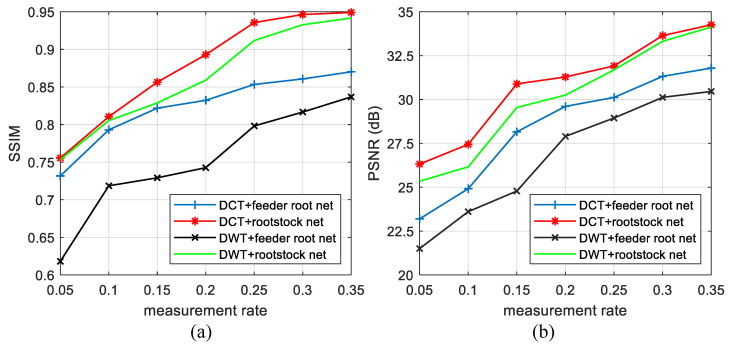
The influence of feeder root branch number on average reconstruction quality in SET11 by (**a**) SSIM and (**b**) PSNR. The branch number is 256.

**Figure 16 entropy-25-01648-f016:**
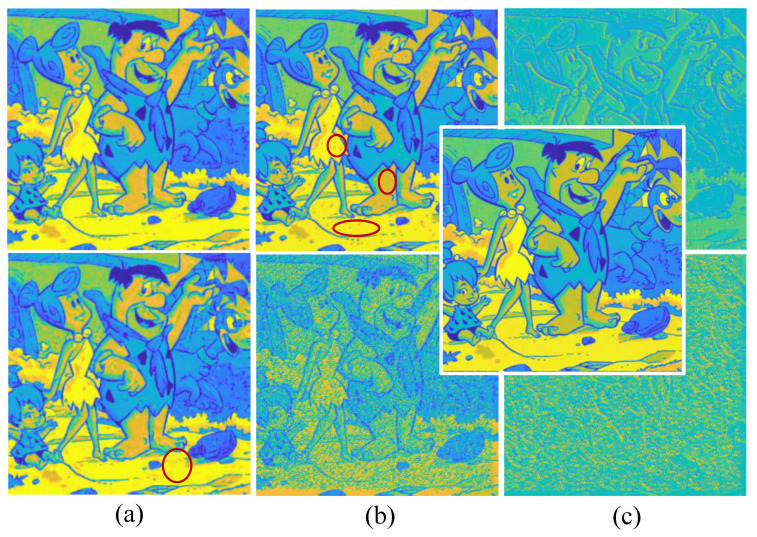
Robustness test results by different error injection rate (1% for the first row, 10% for the second row) for different methods. (**a**) The proposed RootsNet; (**b**) CSNet+; (**c**) AMP-Net. The ground truth results are given in the middle.

**Table 1 entropy-25-01648-t001:** Average reconstruction time in seconds for different methods on SET11.

Time		MR	0.3	0.25	0.2	0.15	0.1

Methods							
OMP [40]			564.3	172.5	58.9	15.6	6.3
IHT [46]			571.8	176.5	57.7	12.5	5.7
SpaRSA [47]			692.3	224.1	71.8	22.6	9.2
OMP-block			99.7	32.9	10.0	2.8	0.9
IHT-block			96.1	30.8	9.3	2.4	0.8
SpaRSA-block			192.4	58.8	18.0	4.9	1.4
ReconNet [13]			0.021	0.022	0.021	0.021	0.021
ISTA-Net+ [20]			0.048	0.048	0.048	0.047	0.048
CSNet+ [16]			0.028	0.027	0.028	0.028	0.028
GPX-ADMM [14]			0.071	0.069	0.070	0.069	0.069
AMP-Net-2BM [26]			0.032	0.031	0.031	0.033	0.031
AMP-Net-9BM [26]			0.041	0.042	0.041	0.041	0.041
**RootsNet**-SinglePC			0.047	0.046	0.046	0.047	0.047
**RootsNet**-Distributed			0.008	0.008	0.008	0.008	0.008

Note: The blue, green, and red colors are the deep-learning-based methods, the worst values, and the best values, respectively.

**Table 2 entropy-25-01648-t002:** Reconstruction quality on SET11.

PSNR/SSIM		MR	0.3	0.25	0.1	0.05	0.01

Methods							
OMP [40]			29.91/0.8641	28.65/0.8517	24.37/0.7143	21.26/0.5646	17.65/0.2426
IHT [46]			29.31/0.8602	28.58/0.8500	24.43/0.7108	21.17/0.5538	17.22/0.2331
SpaRSA [47]			30.86/0.8994	29.42/0.8676	26.12/0.7729	22.13/0.6629	19.17/0.3016
OMP-block			27.14/0.8449	26.48/0.8303	23.60/0.7002	20.03/0.5321	16.895/0.2234
IHT-block			26.66/0.8346	25.21/0.8151	23.52/0.6985	19.65/0.5482	16.01/0.1951
SpaRSA-block			28.23/0.8537	27.70/0.8497	25.42/0.8177	21.72/0.5771	17.62/0.2568
D-AMP [32]			32.64/0.7544	31.62/0.7233	19.87/0.3757	14.38/0.1034	5.58/0.0034
ReconNet [13]			33.17/0.938	32.07/0.9246	27.63/0.8487	21.73/0.6211	17.54/0.4426
DCS [30]			21.98/0.5358	21.85/0.5166	21.53/0.4546	17.67/0.2235	12.51/0.1937
ISTA-Net+ [20]			33.66/0.9330	32.27/0.9127	25.93/0.7840	18.34/0.4715	17.12/0.3251
CSNet+ [16]			33.90/0.9449	32.76/0.9322	27.76/0.8513	21.07/0.6103	20.09/0.5334
GPX-ADMM [14]			33.85/0.9501	32.43/0.9382	26.96/0.8561	19.13/0.5421	18.21/0.4653
AMP-Net-2BM [26]			35.21/0.9530	33.92/0.9417	28.67/0.8654	20.82/0.5614	20.41/0.5539
AMP-Net-9BM [26]			36.03/0.9586	34.63/0.9481	29.40/0.8779	21.88/0.6441	20.20/0.5581
**RootsNet**			34.16/0.9542	32.84/0.9471	28.86/0.8597	24.74/0.7734	22.73/0.7335

Note: The blue, green, and red colors are the deep-learning-based methods, the worst values, and the best values, respectively. ‘PSNR/SSIM’ means the value in front of ‘/’ is PSNR, and the value behind ‘/’ is SSIM.

## Data Availability

Publicly available datasets were analyzed in this study. This data can be found here: https://paperswithcode.com/dataset/set11 (accessed on 18 October 2023).

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
