# Peer review of "A Real-Time and Robust Neural Network Model for Low-Measurement-Rate Compressed-Sensing Image Reconstruction"

_entropy, 2023, doi:10.3390/e25121648_

Round 1

Reviewer 1 Report

Comments and Suggestions for Authors

The proposal presented by the authors is well explained and structured. However, the relevance of the results is not properly justified and there is a lack of experiments in this part of the article.

In the first part of the experimental comparison, there is no analysis of methods using deep learning.

In the second experiment on a general dataset, comparisons are made with methods based on deep learning, but in this dataset the proposal presented cannot be identified as the best option.

The computational cost analysis does not explain the hardware-software platform used in each case, so the results cannot be taken into account.

Only when there are errors can some advantage of the proposed method be identified, but this situation is not identified as a real relevant or likely situation.

Other minor aspects:

Revise the titles of sections 4.2 and 4.2.1.

Revise "materialx

Revise missing spaces

Include reference to Paddlepaddle

Comments on the Quality of English Language

Minor corrections

Author Response

Dear reviewer,

Thank you very much for your time and comments to help with improving the quality of this manuscript. Please find the attached "Revision notes for reviewer 1.pdf".

Reviewer 2 Report

Comments and Suggestions for Authors

The manuscript proposes a neural network, called RootsNet, which integrates CS mechanism in the network to prevent error propagation. The paper is well written and sufficiently clear. Qualitative and some quantitative results carried out in two experiments seem to demonstrate the effectiveness of the proposed approach.

A major insight into the mechanism of the proposed neural network will improve the paper's readability. I suggest making an effort in this sense, maybe exploiting steps in the NN input construction and data handling, and into the learning process.

I suggest a minor revision

Comments on the Quality of English Language

minor changes required, generally good and clear

Author Response

Dear reviewer,

Thank you very much for your time and comments to help with improving the quality of this manuscript. Please find the attached "Revision notes for reviewer2.pdf"

Reviewer 3 Report

Comments and Suggestions for Authors

All the comments are in the attached file. However, in general I think that there are several issues that should be clarified or fixed.

Comments on the Quality of English Language

I have found the reading of the paper quite difficult.

Author Response

Dear reviewer,

Thank you very much for your time and comments to help with improving the quality of this manuscript. Please find the attached "Revision notes for reviewer 3.pdf"

Round 2

Reviewer 1 Report

Comments and Suggestions for Authors

The authors have addressed all my concerns, in particular I believe that the title change is relevant to clearly indicate the main result of the work presented.

Author Response

Thanks for your patience in helping improve the quality of this manuscript.  It's a good training process for us to improve our academic writing abilities.

Reviewer 3 Report

Comments and Suggestions for Authors
